# fMRI-based detection of alertness predicts behavioral response variability

**Sarah E Goodale[1,2]\*, Nafis Ahmed[3], Chong Zhao[3], Jacco A de Zwart[4], Pinar S Özbay[4], Dante Picchioni[4], Jeff Duyn[4], Dario J Englot[1,2,3,5,6], Victoria L Morgan[1,2,5,6], Catie Chang[1,2,3]\***

[1]Department of Biomedical Engineering, Vanderbilt University, Nashville, United States; [2]Vanderbilt University Institute of Imaging Science, Vanderbilt University Medical Center, Nashville, United States; [3]Department of Electrical Engineering and Computer Science, Vanderbilt University, Nashville, United States; [4]Advanced MRI Section, National Institute of Neurological Disorders and Stroke, National Institutes of Health, Bethesda, United States; [5]Department of Neurological Surgery, Vanderbilt University Medical Center, Nashville, United States; [6]Department of Radiology and Radiological Sciences, Vanderbilt University Medical Center, Nashville, United States

**Abstract** Levels of alertness are closely linked with human behavior and cognition. However, while functional magnetic resonance imaging (fMRI) allows for investigating whole-brain dynamics during behavior and task engagement, concurrent measures of alertness (such as EEG or pupillometry) are often unavailable. Here, we extract a continuous, time-resolved marker of alertness from fMRI data alone. We demonstrate that this fMRI alertness marker, calculated in a short pre-stimulus interval, captures trial-to-trial behavioral responses to incoming sensory stimuli. In addition, we find that the prediction of both EEG and behavioral responses during the task may be accomplished using only a small fraction of fMRI voxels. Furthermore, we observe that accounting for alertness appears to increase the statistical detection of task-activated brain areas. These findings have broad implications for augmenting a large body of existing datasets with information about ongoing arousal states, enriching fMRI studies of neural variability in health and disease.

\*For correspondence:
sarah.e.goodale@vanderbilt.edu
(SEG);
catie.chang@vanderbilt.edu (CC)

**Competing interests:** The authors declare that no competing interests exist.

## Introduction

Brain function and behavior vary naturally over time. Behavioral responses can change even across repeated presentations of the same stimulus, and altered levels of neural variability may index individual differences in cognitive function (*Dinstein et al., 2015*; *van Kempen et al., 2019*; *Cohen, 2018*). One major determinant of behavioral and neural variability is the brain's continuously fluctuating level of arousal, referring to the dimension of functional states linked with phenomena including alert wakefulness, drowsiness, and sleep. Growing evidence indicates that levels of arousal closely interact with processes underlying decision-making (*van Kempen et al., 2019*), visuomotor performance (*Makeig et al., 2000*; *Poudel et al., 2018*), perception (*Mather and Sutherland, 2011*), and attention (*Foucher et al., 2004*), motivating incorporation of arousal levels into models of healthy and pathological brain function (*Lee et al., 2014*; *Nassar et al., 2012*; *Englot et al., 2008*; *Motelow et al., 2015*; *Salomon et al., 2011*; *Nashiro and Mather, 2011*; *Jawinski et al., 2019*).

Functional MRI allows for investigating how behavioral variability may be underpinned by the activity in distinct spatial locations, or across large-scale functional networks, in the human brain. For example, variations in attentional (*Rosenberg et al., 2016*), motor, (*Fox et al., 2006*), and

perceptual (*Sadaghiani et al., 2009*) responses, have been linked to fluctuations in spontaneous functional magnetic resonance imaging (fMRI) signals, observed both in focal regions of the brain (*Fox et al., 2006*; *Sadaghiani et al., 2009*; *Hesselmann et al., 2008a*; *Hesselmann et al., 2008b*) and in large-scale network structure (*Rosenberg et al., 2016*; *Sadaghiani et al., 2009*; *Kelly et al., 2008*; *Thompson et al., 2013*; *Sadaghiani et al., 2015*; *Boly et al., 2007*; *Coste et al., 2011*). Temporal variation in spontaneous fMRI signals and connectivity have also been found to be altered in neuropsychiatric disorders as well as in aging (*Yang et al., 2014*; *Nomi et al., 2017*; *Calhoun et al., 2014*). However, relatively few fMRI studies have examined the specific role of dynamically fluctuating arousal states in shaping ongoing neural or behavioral variability (*Dinstein et al., 2015*; *Liu and Falahpour, 2020*). One barrier stems from practical challenges linked with gathering established measures of arousal (such as pupillometry and EEG) during routine fMRI scans – including the need for additional setup time, MRI-compatible hardware, and post-processing for removal of MR-related artifacts. Therefore, the ability to estimate fluctuations in brain arousal from fMRI data itself would broaden the ability to study dynamic interactions between arousal, behavior, and large-scale brain activity. It could also improve the mechanistic interpretation of fMRI analyses, particularly in populations known to have altered arousal and autonomic regulation (*Cohen-Zion and Ancoli-Israel, 2004*; *Hegerl et al., 2016*).

Prior work on the detection of arousal states from fMRI alone has demonstrated that distinct EEG-defined sleep stages may be classified based on fMRI correlation patterns calculated across 1–2 min sliding windows (*Tagliazucchi and Laufs, 2014*; *Tagliazucchi et al., 2012*; *Altmann et al., 2016*). Moreover, it was found that many subjects can begin to lose wakefulness even within the first few minutes of an fMRI scan (*Tagliazucchi and Laufs, 2014*). Focusing on levels of arousal across the waking state, *Wang et al., 2016* identified dynamic connectivity patterns, calculated in 40 s sliding windows, whose expression at each window tracked slow variations in subjects' performance in an auditory vigilance task. In a complementary line of work, we recently developed an approach for extracting continuous fluctuations in arousal from fMRI data alone, this time at the frame-by-frame (TR) temporal resolution of fMRI data (*Chang et al., 2016*). The extracted 'fMRI arousal index' was found to correlate with intracranial electrophysiological arousal and spontaneous eye opening/closing in macaque monkeys (*Chang et al., 2016*), as well as with scalp EEG in human subjects (*Falahpour et al., 2018*). It is possible that the higher temporal resolution of this approach may also provide the opportunity of capturing arousal-dependent behavioral variability during task performance, in a temporally localized manner. If such a measure can indeed track trial-wise neural state changes, it may also open the possibility of incorporating arousal states into analysis of the many task fMRI datasets that lack EEG or pupillometry measures.

To examine this possibility, here we simultaneously record EEG, fMRI, and behavioral responses and examine whether this time-resolved marker of alertness (*Chang et al., 2016*; *Falahpour et al., 2018*) – derived from fMRI data alone – can be harnessed to track trial-to-trial variations in responses to sensory stimuli. We investigate whether an fMRI index of the ongoing arousal state, estimated within a short (5 s) interval before each trial, is predictive of a subject's response to the upcoming stimulus. We also probe the impact of including fMRI-derived arousal covariates in standard statistical analyses of fMRI task activation.

## Results

### fMRI-derived alertness index tracks continuous variations in EEG alertness

In this study, we use an approach for inferring fluctuations in alertness from fMRI data alone, at the temporal resolution of the fMRI scan (*Chang et al., 2016*; *Falahpour et al., 2018*). Briefly, this is carried out by taking the running spatial projection of a spatial map linked with alertness ('template map') onto each successive fMRI volume in the new scan, resulting in a continuous time course of estimated alertness sampled at each fMRI volume (TR). A schematic of this approach is provided in *Figure 1*, with further details in Methods.

Importantly, for unbiased evaluation of the fMRI alertness index, here we derive the template map from a set of *resting-state* EEG–fMRI scans (where subjects rested passively with eyes closed) and apply it to estimate continuous alertness fluctuations in a separate set of *auditory task* scans (in

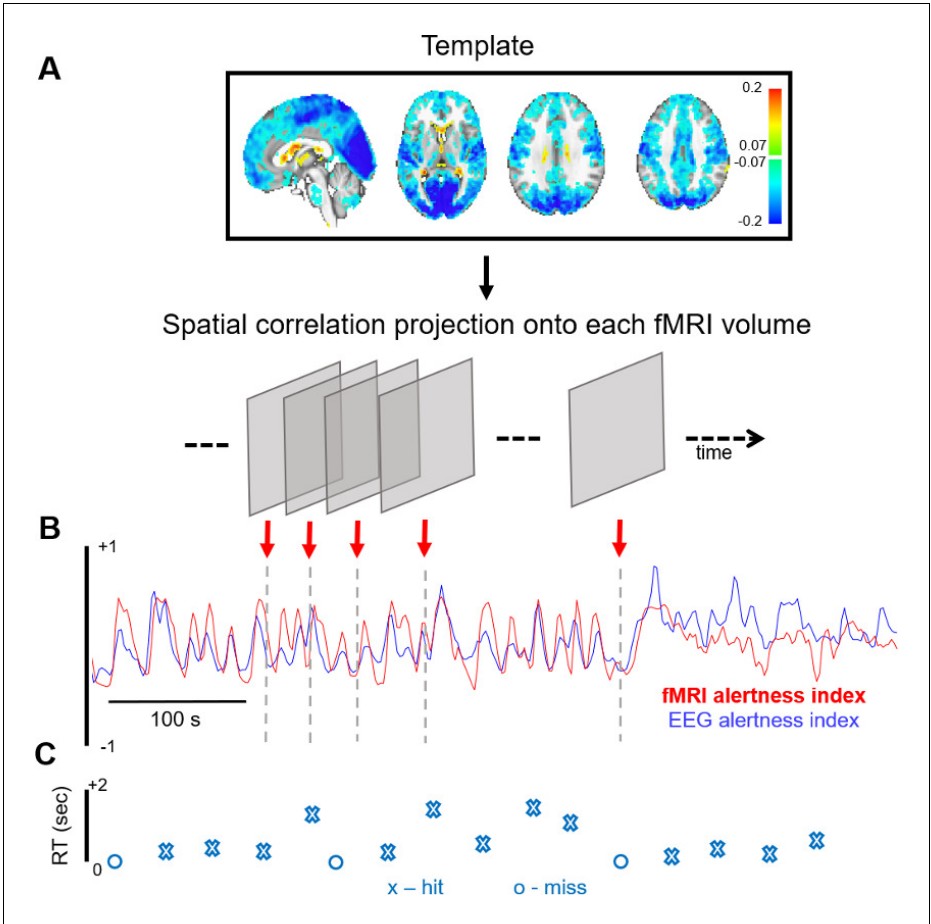

**Figure 1.** fMRI-based inference of alertness fluctuation. (**A**) A spatial template, created a priori (here, using resting-state scans with simultaneous EEG–fMRI), is projected onto each successive volume of a different, auditory task fMRI scan (not used in the template creation) via spatial correlation. These correlation values trace out a time course of estimated arousal fluctuation, which we refer to as the 'fMRI alertness index'. (**B**) To assess the ability to predict electrophysiological alertness in the task scan, the estimated fMRI alertness index (red) is compared with an established EEG vigilance index that was collected simultaneously (the alpha/theta power ratio, convolved with a hemodynamic response function; blue). (**C**) The reaction time (RT) to each auditory stimulus is represented by an 'X' when the subject responded to a given trial and 'O' when they missed.

which subjects were presented with binaural tones at long and unpredictable inter-stimulus intervals, with eyes closed; see Materials and methods). Subjects were requested to press a button as soon as possible upon hearing a tone. For all results in this study, the EEG data from the auditory task scans is only used for validation purposes and does not enter into the creation of the template or the fMRI alertness index.

We first determined how well this 'fMRI alertness index' could track an established index of continuous electrophysiological arousal during the auditory task scans. To quantify this agreement, we calculated the temporal correlation between the estimated fMRI alertness index and a simultaneously measured 'EEG alertness index' across each of the auditory task scans (Materials and methods). *Figure 2A,B* depicts example time courses of the fMRI alertness index (derived independently of these scans' EEG data), superimposed on the measured EEG alertness index for comparison. *Figure 2C* shows the temporal cross-correlation between these two waveforms across the set of task scans, relative to a null distribution that was created by randomly permuting the fMRI alertness index and EEG alertness index across the task scans.

To probe the across-subject reproducibility of fMRI alertness prediction within our dataset, we performed an additional cross-validation analysis in which subjects were divided into two distinct groups. Model training was performed on the resting-state data from one group and evaluated on

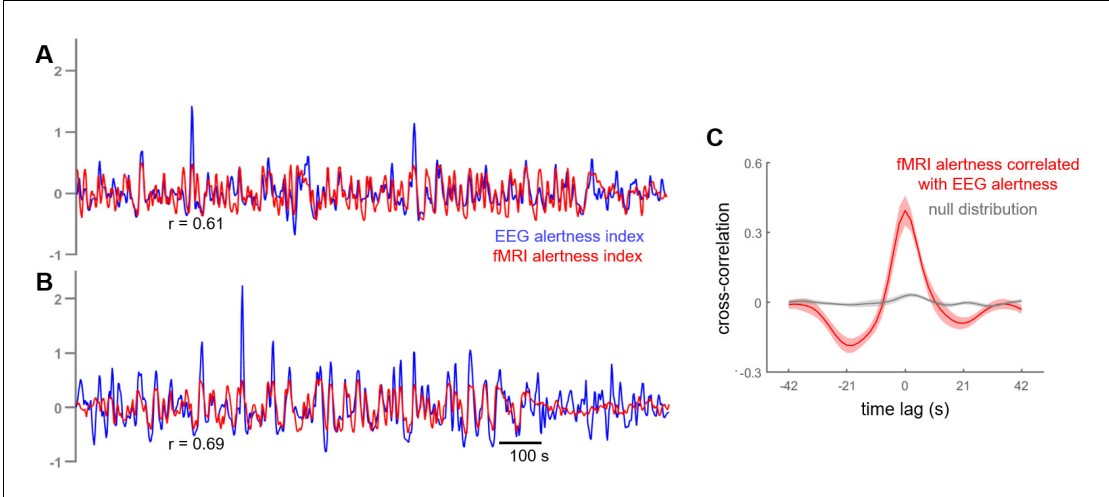

**Figure 2.** Cross-correlation between estimated fMRI alertness index and measured EEG alertness index. (**A, B**) fMRI alertness index (red), superimposed on the EEG alertness index (blue), for two example scans. Note that the EEG data from these scans was not used in the creation of their fMRI alertness time courses, and are only used to evaluate the ability of the fMRI alertness index to track electrophysiological arousal. (**C**) Temporal cross-correlation between the fMRI alertness index and EEG alertness index (mean ± SE, n = 12 scans; red) together with a null distribution (gray) constructed for statistical comparison (see Materials and methods).

The online version of this article includes the following figure supplement(s) for figure 2:

**Figure supplement 1.** Further analysis of the reliability and across-subject generalizability of the alertness estimation approach.

**Figure supplement 2.** Lasso regression model for alertness prediction.

**Figure supplement 3.** Model performance when trained on task data and tested on resting-state data.

the task data of the other group. Cross-correlations between the estimated resting-state fMRI alertness index and the measured EEG alertness index were comparable to the original results (*Figure 2—figure supplement 1*).

## Variations in trial-by-trial response times relate to arousal state and are tracked by an fMRI alertness index

We next aimed to test whether the fMRI alertness index could provide a marker of arousal-dependent behavioral variability in an auditory task. For each scan, all trials were first binned into 'hits' and 'misses', corresponding to whether or not the subject pressed the button in response to the auditory stimulus. The event-locked EEG spectrogram showed clear evidence that hit and miss trials were accompanied by distinct arousal states, with the hit spectrogram showing greater power in the alpha band (8–12 Hz, consistent with increased arousal, particularly at stimulus onset) and the miss spectrogram showing greater power in the lower frequency bands commonly associated with increased drowsiness or sleep (*Figure 3A*).

The time course of the fMRI alertness index closely mirrored the EEG response. Significant differences between hits and misses were observed in both the pre-stimulus (p=3.34e-03, effect size [Cohen's d] = 1.24) and post-stimulus (p=6.40e-06, d = 2.39) intervals, defined, respectively, as the intervals 5 s before, and 10 s after, the stimulus onset (*Figure 3B*; two-sample t-tests). The pronounced post-stimulus peak in the hits trials may again be indicative of stimulus-induced changes in alertness, which manifests with a hemodynamic response delay. Within individual subjects, trials with hits were associated with higher levels of the fMRI alertness index in the pre-stimulus interval compared to trials with misses (*Figure 3C*). Of those subjects who had both hits and misses, all but two followed the expected trend, and the differences between hits and misses were significant (t(8) =3.35, p=0.01, paired t-test; d = 1.11). Our focus was primarily on the *pre-stimulus* interval, as it can more closely represent the internal state of the brain without contribution from changes in arousal induced by the stimulus itself.

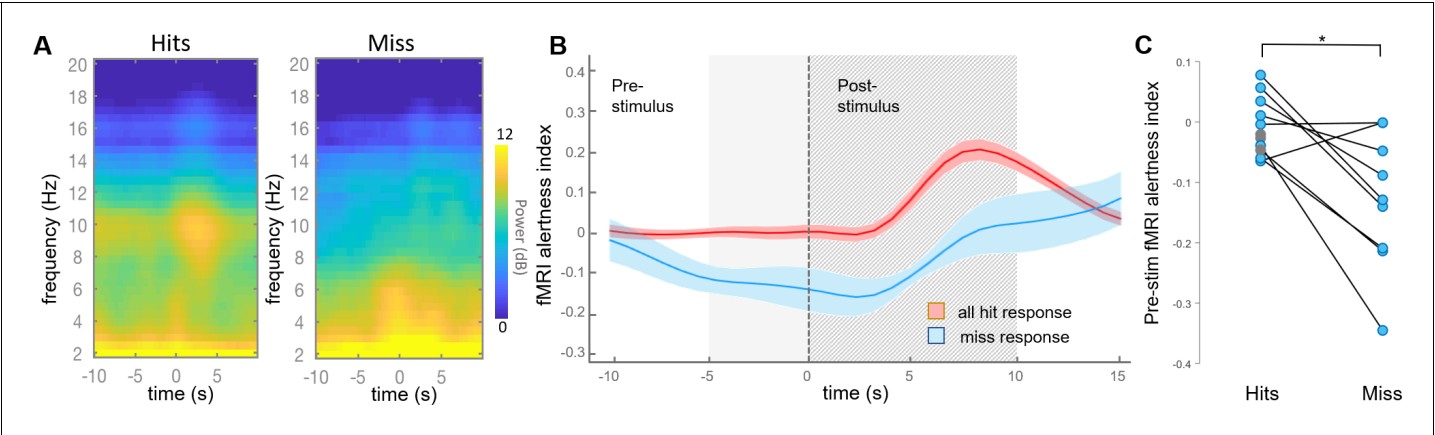

**Figure 3.** fMRI alertness index differentiates between auditory task response (hit) and misses. (**A**) Event-locked EEG spectrogram for hits and misses. During hits, the elevated power in the alpha band may signify higher alertness; during misses, lower levels of alpha and increased power in lower frequencies suggest lower alertness. Time = 0 corresponds to stimulus onset. (**B**) Event-locked fMRI alertness index. Within each subject, the fMRI alertness index surrounding each stimulus was averaged across trials with hits (red) and misses (blue), and the across-subject means are represented by solid lines (shaded area is standard error; n = 12 scans for hits, n = 9 scans for misses). Across subjects, the fMRI alertness index predicted significantly greater levels of alertness in hits compared to misses, both within the pre-stimulus interval (5 s prior to stimulus onset; light gray region) and post-stimulus interval (10 s after stimulus onset; striped gray region). A wider time interval is shown for more complete visualization. (**C**) Visualization of individual subjects' effects. Within individual scans, the fMRI alertness index was averaged within the 5 s pre-stimulus interval of each trial. Scans with only 'hits' are marked as gray circles; no scans had only misses. Within subjects who had both hits and misses, a significant difference in the pre-stimulus fMRI alertness index was found (p=0.01, paired t-test; d = 1.11).

We then further aimed to determine whether the fMRI alertness index could account for variability in reaction times to the upcoming stimulus, in trials with hits. We again focus primarily on the pre-stimulus interval. Here, hits were further binned into an equal number of 'slow' and 'fast' trials, using a reaction-time threshold of 565 ms (median of responses pooled across all subjects). Within the 10 subjects that had both fast hits and slow hits, fast hits were associated with higher levels of the pre-stimulus fMRI alertness index compared to slow hits (t(9)=4.52, p=0.001, paired t-test; Cohen's d = 1.43; *Figure 4A*), and all showed higher mean values during fast compared to slow hit trials. Similar findings were obtained when using subject-specific divisions into fast and slow trials (top and bottom 10%, respectively; *Figure 4—figure supplement 1*). This finding supports the notion that pre-stimulus patterns in spontaneous brain activity encode behaviorally relevant internal states within individual subjects, which may be quantified using an fMRI pattern-recognition approach.

Although our initial focus was on the pre-stimulus interval, we also investigated the modulation of alertness measures following stimulus onset. In the 10 s post-stimulus interval, there was also a main effect of behavioral response for both EEG alpha/theta and the fMRI alertness index (EEG: F(2) = 8.82, p=0.1e-02, fMRI: F(2) = 14.12, p=5.75e-05; *Figure 4C*). Both EEG and fMRI showed post hoc pairwise differences between both fast and slow responses compared to misses, and neither showed a difference between fast and slow hits. A potential explanation is that, while a lower level of alertness before the stimulus led to slower reaction times in 'slow' trials, the incoming auditory stimulus led to a subsequent increase in alertness, approaching that of fast trials. We also examined the trial-by-trial correspondence between the fMRI alertness and EEG alpha/theta power using an offset that accounts for approximate hemodynamic delays (*Figure 4—figure supplement 2*).

Finally, while multi-echo independent component analysis (ICA) has proven effective in reducing head-motion effects (*Kundu et al., 2013*), we also ensured that scans with frame-to-frame head motion exceeding our voxel size (3 mm; corresponding to one resting-state and one task scan; *Supplementary file 3*) did not impact the major findings of this study. Removing these high-motion scans from the analysis had minimal impact on the correspondence between the fMRI alertness index with EEG and behavior (*Figure 4—figure supplement 4*).

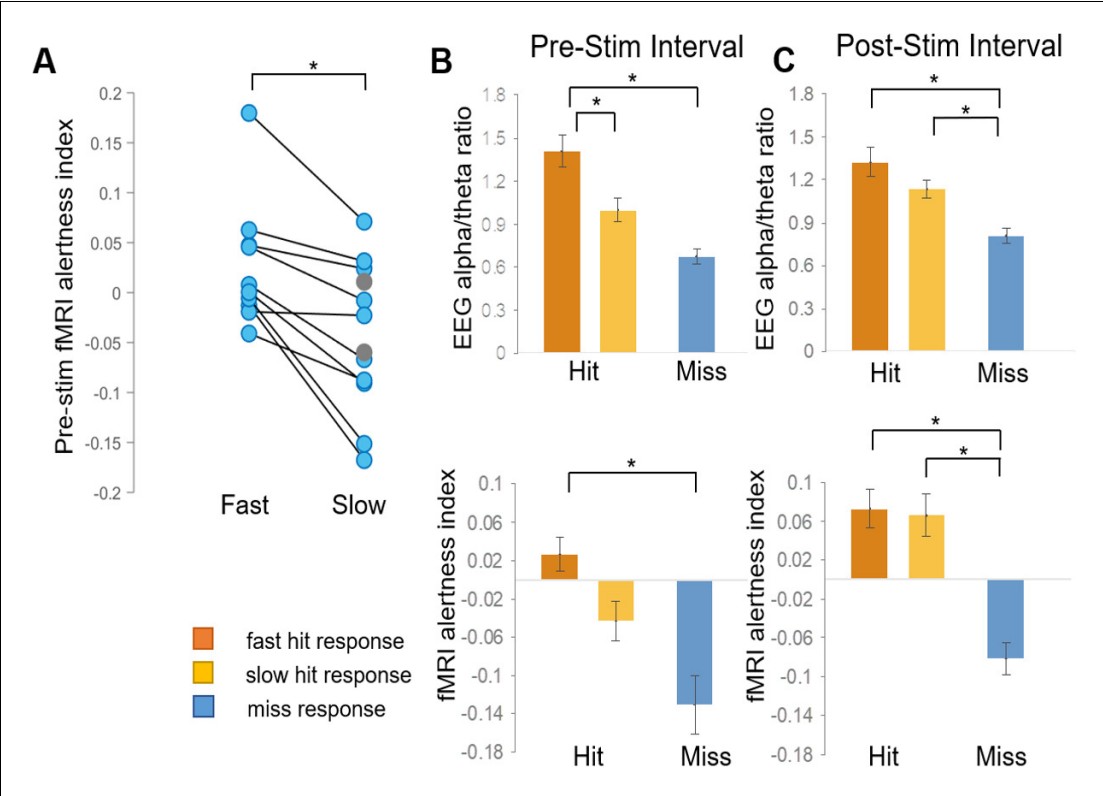

**Figure 4.** fMRI alertness index differentiates between trials with fast and slow reaction times. (**A**) Within individual subjects, the pre-stimulus fMRI alertness index was averaged across fast and slow reaction-time trials. Subjects with only 'slow hits' are marked as gray circles (none had only fast hits). There was a significant difference between fast and slow hits within subjects who exhibited both fast and slow responses (p=0.001, paired *t*-test; d = 1.43). (**B**) Three types of behavioral responses (fast hit, slow hit, and miss) were examined for the fMRI alertness index, as well as the EEG alpha/theta ratio for comparison. In the pre-stimulus interval, both the EEG alpha/theta ratio and the fMRI alertness index showed a main effect of behavioral response across subjects; p<0.05, (one-way ANOVA). Tukey–Kramer post hoc tests, controlled for multiple comparisons, indicated pairwise differences for fast hits (orange) versus slow hits (yellow) for EEG and between fast hits and misses (blue) for both EEG and fMRI (*p<0.05). (**C**) In the post-stimulus interval, a main effect of behavioral response was also found for both EEG and fMRI, and post-hoc tests indicated significant differences in both fast hits and slow hits versus misses (*p<0.05). Error bars represent standard error.

The online version of this article includes the following figure supplement(s) for figure 4:

**Figure supplement 1.** fMRI alertness index differentiates between the 10% fastest and slowest reaction times in each individual subject.

**Figure supplement 2.** Trial-by-trial correspondence between EEG alpha/theta power and fMRI alertness index during pre-stimulus interval.

**Figure supplement 3.** Effects of regressing out low-frequency respiratory and cardiac fluctuations from the fMRI signal.

**Figure supplement 4.** Evaluation of fMRI alertness index with high-motion subjects removed.

**Figure supplement 5.** Reaction time histogram.

## A small number of voxels may suffice for detecting alertness with fMRI

In a previous study of macaque monkeys, we demonstrated that the whole-brain vigilance template could be reduced to its top ~1% of positive and negative voxels without considerable loss in predictive power (*Chang et al., 2016*). In addition, despite the much larger number of negative compared to positive voxels, retaining voxels of both signs was found to be critical (*Chang et al., 2016*). To determine whether a small number of template voxels also suffices for inferring EEG and task response variability in human subjects, we performed a similar experiment, successively thresholding the template to include different percentages of its highest-magnitude positive and negative voxels.

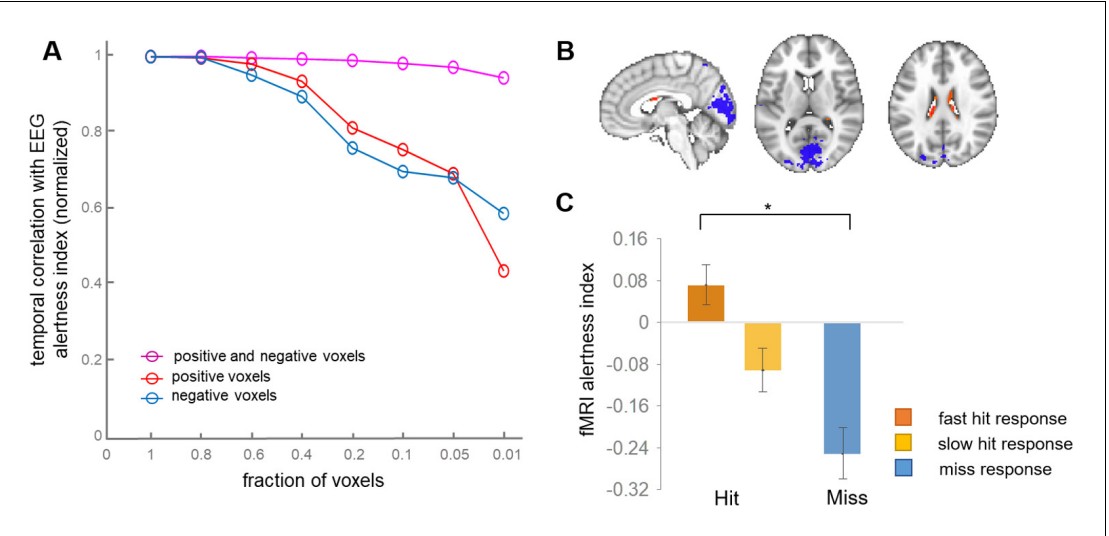

**Figure 5.** Effects of spatially reducing the vigilance template. (**A**) The template was successively reduced to retain the indicated fraction of highest-magnitude voxels drawn from the set of negative, positive, or both negative- and positive-signed voxels. These reduced voxel sets were used as templates for calculating the fMRI alertness index in each scan. The x-axis indicates the fraction of voxels relative to the whole-brain template; that is, at x = 0.2, we used the highest 20% of positive values (red) and the highest 20% of negative voxels (blue), then combined them to make a joint positive and negative template (magenta). To focus on relative effects, all three lines were normalized to their value at the fraction of 1.0. (**B**) Template map produced by retaining only the top 1% of positive and negative voxels of the original whole-brain template, respectively. (**C**) The pre-stimulus fMRI alertness index, generated from this reduced ('1%") template, for fast, slow, and missed trials (see *Figure 4B* for analogous results using the whole-brain template).

We also examined the impact of including only positive- or only negative-valued voxels from the template. When using only positive or only negative voxels, performance was found to decline as smaller fractions of voxels were used (*Figure 5*). By contrast, when jointly including positive and negative voxels, we found that there was little change in the predictive power, even at thresholds down to 1%.

Here, the highest-magnitude (top 1%) voxels with negative sign were localized mainly to primary visual, auditory, and sensory/motor cortices, and those with the highest-magnitude positive sign were found in the ventricles (*Figure 5b*). This highly reduced template map also generated an fMRI alertness index that replicated behavioral results obtained from the whole-brain template map. The fMRI alertness index generated from the reduced template (top 1% positive, top 1% negative voxels), calculated in the pre-stimulus interval, showed a significant main effect of behavioral response (F(2)=10.09, p=0.0005; *Figure 5c*). This finding suggests that despite the broad nature of arousal-related signal changes across the brain, the information is inherently low dimensional, and a small fraction of voxels may suffice for detecting fluctuations in alertness.

## Modeling fluctuations in alertness impacts inference of task activation

Beyond providing information about the brain's ongoing arousal levels, an fMRI-derived alertness index may potentially allow for more reliable detection of task-evoked activation, by offering a means for explicitly modeling state-related fMRI signal changes across an experiment. To examine this possibility, we performed a standard general linear model (GLM) analysis of the auditory task responses (convolved with a canonical hemodynamic response function [HRF]) and quantified the impact of including the fMRI alertness index as a covariate of no interest. For all analyses, a regressor capturing the onset of missed stimuli (convolved with a canonical HRF) was also included as a covariate of no interest. As shown in *Figure 6A*, a model that included only the event-related task responses resulted in weak activation in the expected bilateral auditory and left motor regions, along with negative BOLD signal change in a number of cortical regions, with some resemblance to the pattern of correlations with EEG arousal (*Figure 1*, *Figure 6—figure supplement 1*). Including

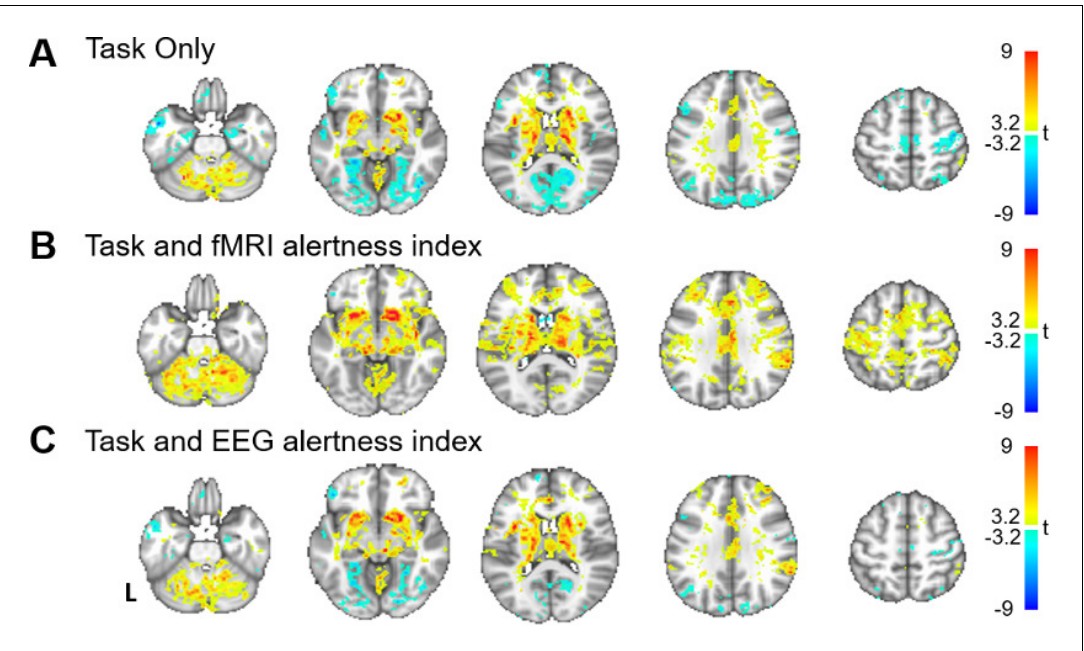

**Figure 6.** Modeling alertness fluctuations in fMRI analysis impacts statistical inference of task-evoked activation. General linear model analyses of responses to event-related auditory task. For each subject, beta maps were calculated for event-related responses to the auditory stimuli, resulting in group-level t-statistics (voxelwise one-sample t-test) for (**A**) a model that includes regressors only for the task stimuli; (**B**) a model that includes task regressors along with the fMRI alertness index as a covariate; and (**C**) a model that includes task regressors along with the EEG alertness index as a covariate.

The online version of this article includes the following figure supplement(s) for figure 6:

**Figure supplement 1.** Comparison between event-related task response and vigilance template.

the fMRI alertness index as a covariate reduced the negative responses and increased t-scores in bilateral auditory cortex, left motor cortex (the region expected to be associated with right-handed button presses), and the contralateral cerebellum (*Figure 6B*). Interestingly, this covariate also markedly increased the activation in areas associated with the 'salience network' (including the anterior cingulate cortex and insula [*Menon and Uddin, 2010*], which are known to co-activate with externally salient or oddball stimuli *Crottaz-Herbette and Menon, 2006*); this may perhaps to be expected given the long and unpredictable inter-stimulus intervals in this task. Using the EEG alertness index as a regressor also reduced the negative activation and increased the t-statistics in auditory, motor, and salience areas (*Figure 6C*), though surprisingly, to a somewhat lesser extent than with the fMRI index.

### Comparison to lasso regression model

Finally, to investigate how this template-based method compares against a more conventional temporal regression approach, we also constructed lasso regression models (see Materials and methods for further details). The lasso model yielded correlations with the EEG alertness index that were similar to, though slightly lower than, those generated by the template-based approach (*Figure 2—figure supplement 2*). Furthermore, the lasso model that was trained on fMRI time series from a 268-region atlas (*Shen et al., 2013*) was superior to that which used the time series from all voxels as input.

## Discussion

The present results indicate that ongoing variations in subjects' alertness, as directly estimated from fMRI data alone, may track both concurrently measured EEG and inter-trial behavioral responses.

These findings are in line with a growing literature indicating that time-varying, internal state changes may be decoded based on patterns in fMRI data, which may be harnessed to study specific dimensions of cognitive and physiological variability such as arousal, attention, and emotion (*Rosenberg et al., 2016*; *Thompson et al., 2013*; *Chang et al., 2015*). In addition, the ability to estimate vigilance during fMRI tasks allows for modeling state-dependent variability in task performance (*Podvalny et al., 2019*) and BOLD responses, which in turn may improve the reproducibility and interpretation of fMRI datasets that lack EEG or other external vigilance measures.

Changes in arousal are coordinated by ascending projections from the brainstem, via regions including thalamus, basal forebrain regions, and posterior hypothalamus (*Jones, 2005*; *Saper et al., 2005*), and manifest in altered responsiveness to sensory stimuli (*Ogilvie, 2001*). During periods of drowsiness and light sleep, large-amplitude and extensive fMRI signal changes have been observed in which much of the cortex is modulated together, and certain subcortical structures (including the thalamus) display opposing fluctuations (*Chang et al., 2016*; *Falahpour et al., 2018*; *Olbrich et al., 2009*; *Goldman et al., 2002*; *Moosmann et al., 2003*; *Ong et al., 2015*; *Poudel et al., 2014*; *Liu et al., 2012*; *Feige et al., 2005*). These inversely related signals may be the hemodynamic correlates of subcortical arousal modulation, and perhaps reflect inhibition of incoming sensory signals to the cortex by the thalamus in a feedback role (*Sherman and Guillery, 2002*). Here, we use the temporal expression of an EEG arousal-linked fMRI signal pattern ('template') to estimate a subject's alertness fluctuation across the course of an fMRI scan. In addition to finding that such an fMRI-based inference of alertness correlates with electrophysiological arousal, consistent with prior work (*Chang et al., 2016*; *Falahpour et al., 2018*), the use of binaural tones here enabled us to investigate its bidirectional interaction with sensory stimuli.

In the pre-stimulus interval, the fMRI alertness index was indicative of upcoming behavioral responses. This finding extends a body of prior work linking fMRI spontaneous activity with behavior and perception (e.g., *Sadaghiani et al., 2009*; *Kelly et al., 2008*; *Thompson et al., 2013*; *Sadaghiani et al., 2015*), and it is also interesting to consider the potential links between the present findings and large-scale network patterns that have been previously shown to predict vigilance or tonic alertness. In particular, higher levels of pre-stimulus activity in the default-mode network and/or salience (cingulo-opercular) network, and lower levels in task-positive/dorsal attention networks, have been associated with auditory stimulus detection (*Sadaghiani et al., 2009*) and faster reaction times in visual psychomotor vigilance task (*Thompson et al., 2013*). In line with these observations, our template map – derived from EEG spectral power fluctuations – is negative in areas spanning dorsal attention network and had weak positive correlations in the anterior cingulate cortex, a key node of the salience network; however, we did not see group-level positive effects in the default-mode network. Future work may investigate the degree to which shared and distinct physiological mechanisms account for these observations, as well as the extent to which arousal interacts with other neural sources of behavioral variability, such as mind wandering (*Kucyi et al., 2016*).

Given that broad regions of cortex together exhibited correlated negative responses to arousal, we also investigated whether a much smaller set of voxels may suffice for extracting an alertness measure from fMRI. Indeed, we found that retaining even just 1% of positive voxels and 1% of the negative voxels produced only a 5.7% decrease in performance on average (in terms of correlation with EEG alertness). In addition, the pre-stimulus alertness index – estimated from this highly reduced template – replicated the main effect of behavioral response seen with that estimated from the whole-brain template. This effect suggests that a joint pattern of positive and negative voxels may be most critical for decoding arousal with the present approach, consistent with our prior results in macaque monkeys (*Chang et al., 2016*). However, unlike in the macaque study, where the strongest positive voxels in the template (i.e., those which had the greatest positive correlation with alertness) were localized to thalamus, here we found the strongest positive correlations in cerebrospinal fluid (CSF), though weaker positive clusters were also found in mediodorsal thalamus and anterior cingulate cortex. One potential difference may come from the manner in which the template map was derived – here from the EEG alpha/theta ratio during eyes closed, and in the macaque study from spontaneous eye opening/closure. Positive correlations between an arousal index and CSF have been seen in prior human EEG–fMRI studies that use a continuous EEG power ratio (alpha to theta and delta bands *Falahpour et al., 2018*) and discrete EEG-defined vigilance staging (*Olbrich et al., 2009*), where ventricular signals in the latter were suggested to potentially relate to effects of heart rate (HR) on fMRI data (*de Munck et al., 2008*). A pattern very similar to the reduced

template, with opposing gray-matter and ventricular signals, was also reported to comprise the first principal component of the global fMRI signal that is modulated by sleep depth (*McAvoy et al., 2019*). An inverse relationship between CSF and gray-matter signals has also been found to correlate with EEG (*Fultz et al., 2019*), and both EEG and autonomic responses (*Özbay et al., 2019*), during sleep. One mechanistic explanation offered recently is that widespread fMRI signal changes, generated by either neuronal or autonomic activity, lead to antagonistic fMRI signal changes in the downstream venous vasculature, in part due to a disjoint timing and amplitude relationship of blood volume and blood oxygenation effects (*Özbay et al., 2018*). Further work will be necessary to establish potential links between these observations and those of studies which have observed CSF fMRI signals in other EEG frequencies (*Kiviniemi et al., 2016*).

A general challenge in understanding the effects of arousal in fMRI stems from the complex assortment of neural, metabolic, and physiological factors contributing to BOLD fMRI signals (*Liu, 2016*; *Duyn et al., 2020*). Fluctuations in arousal are closely coupled with physiological changes, such as changes in HR and respiratory patterns (*Özbay et al., 2019*; *Yuan et al., 2013*). Therefore, it is hard to disentangle these various effects with certainty. However, when the goal is to detect vigilance changes from fMRI, systemic physiological effects in fMRI signals may, in fact, contribute useful state-related information. In additional experiments, we found that projecting out low-frequency fluctuations in HR and RV generally reduced the correlation between the fMRI and EEG alertness measures (*Figure 4—figure supplement 3* and Materials and methods), suggesting that retaining low-frequency systemic physiological components may present useful information when the goal is to extract arousal fluctuations from the fMRI data. The whole-brain template derived here also overlaps very closely with a spatial pattern linked with sympathetic arousal in stage-2 sleep (*Özbay et al., 2019*; *Özbay et al., 2018*) as well as in drowsy wakefulness and stage-1 sleep (*Chang et al., 2018*), whereby gray-matter regions (predominantly primary sensory cortex) are anti-correlated with BOLD signal in deep white matter and CSF. Our template map may, therefore, reflect in part the concurrent modulation of central and peripheral arousal, and their systemic physiological correlates.

Beyond providing information about ongoing arousal levels, an fMRI-derived alertness index may also allow for more precisely identifying task-evoked activation. While it is widely recognized that natural drifts in wakefulness are common during fMRI, few studies explicitly model dynamic arousal fluctuations in standard fMRI task analyses, as concurrent EEG and pupillometry is relatively uncommon during routine fMRI scans. Here, we observed that including the fMRI alertness index as a covariate increased the statistical significance of task-active sensory and motor regions as well as canonical salience network regions (*Menon and Uddin, 2010*; *Crottaz-Herbette and Menon, 2006*), and reduced widespread cortical deactivations. Regarding the latter, it is also worth noting that when only the task stimuli are modeled, the resulting activation map bears some resemblance to the vigilance template itself, showing similar patterns of negative BOLD signal changes and positive signals in white matter (*Figure 6—figure supplement 1*). This phenomenon might be explained by task-induced arousal and autonomic fluctuations, which are mitigated when including an arousal covariate. However, an important caveat is that the 'true' activation state of the brain is not known; for instance, here one cannot be certain whether diminishing the negative responses with an alertness covariate is also removing local neural effects. We also note that this analysis was not intended to suggest an optimal set of nuisance covariates, but to provide an indication of how covarying for measures of alertness may impact fMRI analyses. Future work may investigate this phenomenon in a wider range of task contexts.

A major goal of template-based alertness estimation is to provide an avenue for researchers to infer fluctuation in alertness from fMRI data when external measures such as EEG and pupillometry are not available in the scanner, which is the case for most existing fMRI data and publicly shared databases. Given a template map trained on a set of EEG–fMRI data (such as this one), researchers could apply it to other fMRI datasets to extract time courses of alertness fluctuation. The present results suggest that, in addition to correlating with an electrophysiological alertness index, this fMRI alertness index may be sensitive to brain states that modulate behavioral responses. Nonetheless, the approach currently has several limitations. One is that it currently detects primarily *relative*, rather than absolute, levels of alertness across a scan. Since spatial correlation is used to determine a frame-by-frame alertness measure, the fMRI alertness index is restricted between −1 and 1. In addition, as fMRI signals are often detrended to remove scanner artifacts, such an index may not

detect slower baseline shifts in alertness. In addition, we and others observe inter-subject and inter-scan differences in the performance of fMRI alertness estimation (*Falahpour et al., 2018*). Such differences may partially reflect the degree of alertness fluctuation present in the fMRI data itself (with greater modulation of alertness lending itself to better prediction), but these questions are currently under investigation.

One key limitation of this study is the small sample size. As a result, the effect sizes of the current results may be overestimated, and findings may not generalize well to the population from which they are sampled. Future work should be carried out with this approach using larger sample sizes as well as other experimental conditions. Nonetheless, there are several factors that increase our confidence in the present results: our scans are long (24.5 min per scan), and the replication analyses carried out in *Figure 2—figure supplement 1* and *Figure 4—figure supplements 1* and *4* further support the generalizability of this approach. Furthermore, although the present study centered on predicting task-related behavioral responses, we also performed an analysis in which the template was constructed from the task scans, and the fMRI alertness index was evaluated on the held-out resting-state scans, obtaining comparable results (*Figure 2—figure supplement 3*). Another limitation is that here, we only considered a template-based approach and a lasso regression model to predict alertness. It is possible that other approaches could be used to extract dynamic measures from alertness from fMRI. However, our current results with these two models support the notion that vigilance can be robustly estimated from fMRI data, and indicate that there may be multiple approaches for extracting this information. Further investigation of alternate linear models for vigilance prediction would be an interesting area to develop in future work, including the use of other regularization methods as well as parcellation atlases for deriving input time series.

In conclusion, our results indicate that time-resolved predictions of alertness, directly estimated in fMRI data without the need for simultaneous EEG, may capture inter-trial behavioral responses and continuous internal state variations. These findings have implications for broadening the study of brain arousal, and its interactions with cognitive and perceptual variability, in the large number of fMRI studies that lack external vigilance measures.

## Materials and methods

### Subjects and data acquisition

Simultaneous EEG–fMRI data were acquired from 14 healthy, right-handed adult subjects (eight females, aged 26 ± 4 years). All subjects provided written informed consent, and human subjects protocols were approved by the Institutional Review Boards of the National Institutes of Health and Vanderbilt University. From these subjects, scans acquired in the eyes-closed resting state and/or with a sparse auditory task (described below) were considered for analysis. One scan was excluded due to a buffer overflow error in the EEG data acquisition. This yielded a total of 23 scans (11 resting-state scans and 12 auditory task scans), each lasting 24.5 min. Further details about the scans corresponding to each subject are provided in *Supplementary file 1*.

MRI data were acquired on a 3T Siemens Prisma scanner (Siemens, Erlangen, Germany) with a Siemens 64-channel head/neck coil. A high-resolution, T1-weighted structural image (TR = 2200 ms, TE = 4.25 ms, flip angle = 9 deg, inversion time = 1000 ms, matrix = 256×256, 160 sagittal slices, 1 mm isotropic) was acquired for anatomic reference. BOLD fMRI data were acquired with a multi-echo, gradient-echo EPI sequence, with the following parameters: flip angle = 75 deg, TR = 2100 ms, echo times = 13.0, 29.4, and 45.7 ms, voxel size = $3 \times 3 \times 3$ mm$^3$, slice gap = 1 mm, matrix size = 82×50, 30 axial slices, acceleration factor = 2. The duration of each scan was 24.5 min (700 volumes). Scalp EEG was acquired simultaneously with fMRI using a 32-channel MR-compatible system (BrainAmps MR, Brain Products GmbH) at a sampling rate of 5 kHz and was synchronized to the scanner's 10 MHz clock to facilitate reduction of MR gradient artifacts. EEG channels were referenced to FCz. Photoplethysmography (PPG) and respiration belt signals were also acquired during the scans (Biopac, Goleta, CA). The PPG transducer was placed on the left index finger, and MRI scanner (slice) triggers were recorded together with the physiological and EEG signals for data synchronization.

For the resting-state scans, subjects were instructed to keep their eyes closed and to stay awake as best as possible. For the auditory task, binaural tones were delivered (VisuaStim Digital;

Resonance Technology, Northridge, CA) with randomized inter-stimulus intervals. For the auditory task, data were drawn from scans having either of two versions of this task, where the two versions differed only in the timing of tone delivery. In one version (corresponding to half of the scans), the inter-stimulus interval (ISI) ranged between 29 and 41 s (41 trials per scan); in the other version, tones were presented with ISI ranging from 8 to 88 s, from which we retained only those which occurred at least 24 s after the previous stimulus (and at least 10 s prior to the subsequent stimulus), resulting in 29 trials per scan. Subjects were asked to make a right-handed button press as soon as possible upon hearing a tone (*Lim and Dinges, 2008*). Auditory tones were delivered via earbuds placed over subjects' ear plugs. During the task scans, subjects were also instructed to keep eyes closed. One rationale for using an eyes-closed auditory task (e.g., as opposed to an eyes-open or visual task) is that it avoids the possibility of subjects missing a stimulus due to variation in their gaze.

Although the instruction was for subjects to stay awake, the long scans were designed to elicit variability in vigilance state – particularly, reduced wakefulness and drifts between wakefulness and light sleep. A random half of the resting-state and task data were independently sleep staged by two certified technologists at the Vanderbilt University Medical Center. Standard sleep scoring was carried out in 30 s epochs, according to standard AASM criteria (*American Academy of Sleep Medicine, 2007*). These ratings indicated that, across all scans combined, 85.6% of epochs were rated as 'wake', suggesting that subjects were mostly alert. It is therefore likely state sleep was not a major factor in our experiments.

## fMRI and EEG pre-processing

Within each scan, the time series of 700 volumes corresponding to each of the three echo times were processed in the following way. For all three of these echo time series, the first seven volumes were dropped to allow magnetization to reach steady state. Then, motion coregistration (six-parameter rigid body alignment) and slice-timing correction were carried out using the functions *3dvolreg* and *3dTshift* in AFNI (https://afni.nimh.nih.gov/afni). For motion coregistration, the alignment parameters were estimated only for the time series of the second (middle) echo, and the resulting parameters were applied to the time series of all three echoes. Maximum frame-to-frame displacement was also calculated, as shown in *Supplementary file 3*. Following this initial processing, multi-echo ICA denoising was carried out using *tedana* 0.0.9a (*Elizabeth and Salo, 2020*; *Kundu et al., 2012*; *Kundu et al., 2013*). Briefly, this procedure aims to separate fMRI signal components generated from BOLD mechanisms from those arising from non-BOLD signal sources (such as scanner drifts and head motion). Spatial ICA is performed on the multi-echo fMRI data, and the resulting components are sorted using automated criteria to determine those which exhibit linear scaling in their percent-signal-change as a function of echo time ('BOLD' components) and those which do not ('non-BOLD' components). The data are then re-constructed using only the former. A more complete description of this approach can be found in *Kundu et al., 2012*, and several studies have demonstrated its efficacy in reducing artifacts including head motion, and aliased cardiac pulsatility and breath-to-breath respiratory artifacts (*Kundu et al., 2013*), the latter of which are otherwise handled with methods such as RETROICOR (*Glover et al., 2000*). However, fMRI variance due to low-frequency respiratory volume (RV; *Birn et al., 2006*) and cardiac rate (*Shmueli et al., 2007*) fluctuations are unlikely to be removed by this procedure, as they arise from a BOLD mechanism (*Kundu et al., 2012*). Here, we calculated RV and cardiac rate signals but did not initially project them out of the data, as their impact was investigated in the analysis (see Methods). Following multi-echo denoising, fMRI data were nonlinearly registered to a standard-space MNI152 template using SPM (https://www.fil.ion.ucl.ac.uk/spm/), followed by fourth-order polynomial detrending and spatial smoothing (FWHM 3 mm) in AFNI.

For the EEG data, reduction of gradient and ballistocardiogram (BCG) artifacts was carried out using BrainVision Analyzer 2 (Brain Products, Munich, Germany), with parameters and procedures described in *Moehlman et al., 2019*. Briefly, gradient artifact reduction followed the average artifact subtraction technique (*Allen et al., 2000*) using slice triggers. Following gradient artifact correction, EEG data were downsampled to 250 Hz. BCG artifact correction proceeded in two steps. First, an artifact template locked to cardiac R-peaks was subtracted from the data, after accounting for an estimated temporal offset between the R-peak and the BCG artifact. Second, independent component analysis (ICA) was performed on the template-subtracted data, and components likely related

to residual BCG were removed by re-constructing the data (via inverse ICA) without the artifactual components. Putative BCG artifact components were manually determined based on the presence of temporal deflections locked with the cardiac cycle, a spatial topography consistent with BCG artifact, and a relatively large contribution to the global field power. For this study, we removed at most two components per scan, to avoid removing potential signals of interest.

## EEG-based alertness index

A measure of alertness was computed from the EEG signal (averaged over channels P3, P4, Pz, O1, O2, Oz) within the 2.1 s interval of each fMRI time point (TR) by taking the ratio of the root mean square (rms) amplitude in the 8–12 Hz range over the rms amplitude in the 3–7 Hz range (alpha/theta ratio). Various EEG-based metrics have been associated with wakefulness; however, many are related to the ratio of power in middle frequency bands (i.e., alpha, beta) to the power in lower frequency bands (i.e., delta, theta) (*Olbrich et al., 2009*; *Klimesch, 1999*; *Oken et al., 2006*; *Jobert et al., 1994*; *Wong et al., 2013*; *Horovitz et al., 2008*). The alpha/theta ratio has previously been used in several human EEG–fMRI studies (*Horovitz et al., 2008*; *Laufs et al., 2006*).

The EEG alpha/theta time course was temporally aligned to the fMRI time course by removing the first seven timepoints. For analyses that directly correlate EEG with the fMRI alertness index (*Figures 1*, *2*, *5*, and *6*), the aligned EEG alpha/theta time course was mean-centered and convolved with the default gamma-variate HRF provided in SPM (https://www.fil.ion.ucl.ac.uk/spm/) to account for approximate hemodynamic delays in relating EEG to fMRI, and band-pass filtered with nominal cutoffs from 0.01 to 0.2 Hz (using the *bandpass* function in MATLAB) to approximate the bandwidth of the fMRI data. The resulting signal is referred to as the 'EEG alertness index'. For the analyses in *Figure 4*, *Figure 4—figure supplement 2*, we use the raw (not hemodynamically filtered) version of the EEG alpha/theta ratio.

## fMRI-based alertness index

A template-based fMRI-based estimate of alertness was calculated according to the approach in *Chang et al., 2016*; *Falahpour et al., 2018*; *Figure 1*. Briefly, this approach centers on the use of a voxelwise spatial pattern ('template' map), indicating the degree to which a given voxel increases or decreases its activity in concert with fluctuations in alertness, as measured using EEG. Given such a template map (derived from a set of scans where concurrent EEG is available), one may then apply it to derive a time course of alertness in a new scan using the fMRI data alone (i.e., even if that new scan does not contain simultaneous EEG). To allow for unbiased estimates of performance, the template map was constructed using the set of resting-state scans, and its performance was analyzed in the set of auditory task scans. To create the template, the preprocessed fMRI data from each subject's *resting-state* scan was temporally correlated, on a voxelwise basis, with the subject's corresponding EEG alertness index, resulting in a voxelwise spatial map of Pearson's correlation coefficients. The resulting correlation maps underwent a Fisher-Z transformation, and were averaged across subjects to create a single, average template map ('vigilance template'). Given this group-average vigilance template (3D spatial map), a time course of alertness ('fMRI alertness index') was estimated for each *auditory task* scan's fMRI data without using the EEG, as auditory task scans were not used in forming the template map. After temporal z-scoring, a time course of alertness ('fMRI alertness index') was estimated for each task scan by projecting the vigilance template, on a volume-by-volume (TR-by-TR) basis, onto each volume of the task scan. To validate the correspondence between the fMRI alertness index and the EEG alertness index on the task data, we calculated the temporal cross-correlation between these two waveforms (*Figure 2A,B*). A null distribution was generated by taking 1000 random permutations (by shuffling the assignment of EEG and fMRI alertness time courses between subjects) across the task scans. Their collective means and standard errors were compared in *Figure 2C*.

To further investigate the reproducibility of fMRI alertness prediction, subjects were divided into two non-overlapping subgroups for both resting and task conditions (*Supplementary file 2*). We then created a template from the resting-state data of one group and applied it to the task data of the other (non-overlapping) group (*Figure 2—figure supplement 1*). Furthermore, to test whether a model trained on task data can be used to predict vigilance in resting state, we also performed the

reverse, constructing our template this time with the task scans from each subject group, and evaluating the fMRI alertness index on the held-out resting-state scans (*Figure 2—figure supplement 3*).

## Relationship between EEG/fMRI alertness and behavioral responses

To investigate the fMRI alertness index (and its comparison to EEG) in relation to trial-by-trial behavioral responses, we extracted segments of these waveforms in the [−5 s to 10 s] interval around each stimulus onset. To achieve more precise alignment with the stimulus onsets, these time courses were first interpolated 2.5-fold to a 0.84 s temporal grid to obtain a finer sampling axis. Each segment was then sorted according to whether it was a 'hit' (button press response) or a 'miss' (no button press response or a response of >5 s). This resulted in a total of 310 hits and 90 misses, pooled across scans. The 'hits' were further separated based on the subject's reaction time into a slow hit (>565 ms, n = 155) and a fast hit (<565 ms, n = 155), where the threshold was defined by the median across all subjects' pooled trials. This threshold is close to the 500 ms value commonly used to define performance lapses (*Lim and Dinges, 2008*).

For the time courses shown in *Figure 3B*, temporal segments of the fMRI alertness index corresponding to 'hits' were first averaged across trials within each subject, and the mean and standard error across subjects at each time point are depicted by the shaded error bars. The same was repeated for 'misses'. For further analyses across behavioral responses (*Figure 3C*, *Figure 4*), time points of the fMRI (or EEG) data segments for each trial were averaged within the indicated interval of interest (e.g., 'pre-stimulus'); then, within each scan, these values were averaged across trials of a given type (e.g., 'miss'), resulting in one value per scan (per trial type). Comparison between trial types within the pre-stimulus interval were carried out using paired t-tests. Modulation of the fMRI alertness index (or EEG alpha/theta power) across fast, slow, and missed responses was assessed using a one-way ANOVA, with post hoc multiple comparisons analyses using the Tukey–Kramer method. Cohen's d was used to assess effect sizes. An additional analysis was performed investigating the use of intra-individual threshold divisions as a comparison, considering the 10% fastest and slowest trials for each subject (*Figure 4—figure supplement 2*). A histogram of all subjects' reaction times is provided in *Figure 4—figure supplement 5*.

## Template thresholding experiment

Our prior work indicates that only a small fraction of the spatial template may be necessary for accurate vigilance prediction in macaques (Figure S9 in *Chang et al., 2016*). Based on these findings, we wanted to observe the impact of thresholding the template to smaller and smaller numbers of voxels in human subjects. This was performed sequentially from the full template down to 1% of the voxels with the strongest positive and negative correlation with the fMRI alertness index. Starting from the full template map (*Figure 1*), at each iteration a new template was created by reducing the voxel sets to the fraction of highest positive and negative correlations from the following percentages: 100%, 80%, 60%, 40%, 20%, 10%, 5%, and 1%. The fMRI alertness index derived from the reduced voxel template was compared via temporal cross-correlation over all time points to the EEG alertness index for all subjects, over a set of lags from −10 to 10 TRs (*Figure 5*). This process was repeated using only the top percentage of negative-valued voxels in the template following the same sequence of thresholds, as well as again with the top percentages of positive-valued voxels.

## General linear model analysis of auditory-stimulus-evoked responses

To test whether explicitly modeling alertness fluctuations could improve the statistical inference of task-evoked activation, we performed a standard GLM analysis on the auditory stimulus-evoked responses. We tested three models: the first model contained only the task stimuli (where trials with and without responses were separated into two regressors), the second model included task regressors along with the EEG alertness index, and the third model contained the task regressors and the fMRI alertness index as regressors. For each model, we calculated the beta map for the regressor corresponding to trials that had responses and applied a voxelwise one-sample t-test across scans (*Figure 6*).

## Impact of low-frequency physiological signals on fMRI alertness prediction

Given that changes in alertness are accompanied by changes in systemic physiological processes such as breathing and HR (*Özbay et al., 2019*; *Yuan et al., 2013*), we also asked how the prediction of EEG alertness from fMRI would be impacted if low-frequency respiration and cardiac effects were first removed from the fMRI data. We hypothesized that if changes in breathing and HR strongly co-vary with alertness, removing these components from fMRI data might adversely impact the inference of alertness from fMRI.

To examine this question, we re-constructed the template map and the calculated fMRI alertness index after regressing out RV and HR from all fMRI data. Specifically, following procedures in *Chen et al., 2020*; *Chang and Glover, 2009*, an RV time course was calculated as the standard deviation in a 6 s sliding window of the respiration belt waveform, and an HR time course was calculated as the (inverse of the) mean inter-beat-interval from the PPG waveform, again in a 6 s window around each TR. The resulting signals were zero-meaned and convolved with respiration (*Birn et al., 2008*) and cardiac (*Chang et al., 2009*) response functions, respectively, along with their temporal and dispersion derivatives, and band-pass filtered from 0.01 to 0.2 Hz. The basis set was then regressed out of the fMRI signal at each voxel.

To quantify the impact on fMRI alertness estimation, the maximum temporal cross-correlation was calculated between the fMRI alertness index and the EEG alertness index before and after physiological removal (*Figure 4—figure supplement 3A*). To assess the degree to which the fMRI alertness index is modulated with behavioral response after this physiological regression step, we performed the same statistical analysis on the 'fast hit', 'slow hit', and 'miss' response types (*Figure 4—figure supplement 3B*). The correspondence between the fMRI alertness index and the EEG alertness index was reduced in most subjects following physiological signal removal. Nonetheless, the fMRI alertness index exhibited similar trial-wise modulation as the original data.

## Lasso-based fMRI alertness index

While our primary analyses used a spatial template-based prediction approach, we also examine an alternate, temporal (lasso) regression model. Just as with the template method, here we train the lasso model on the resting-state fMRI scans and evaluate the performance on the task scans. Two models were considered: one that uses the time courses of all fMRI voxels as input to the model and another that uses ROI time courses extracted from 268 regions of interest (*Shen et al., 2013*) as an initial dimensionality reduction step. For each approach, the resting-state scans were first temporally concatenated across subjects. We then fit a lasso model to this concatenated resting-state time series, using 10-fold cross-validation to choose the regularization parameter (lambda). We then applied the resulting lasso coefficients to the task data, to estimate an alertness time course in each task scan (*Figure 2—figure supplement 2*).

## Statistics and software

All statistical tests are two tailed. Details of the statistical methods are provided in the sections 'Relationship between EEG/fMRI alertness and behavioral responses' and 'General linear model analysis of auditory-stimulus-evoked responses' above. All analyses were performed in MATLAB (R2018b).

## Acknowledgements

This work was supported by NIH grants K22 ES028048 (CC), R01 NS112252 (DJE, CC), R01 NS108445 and R01 NS110130 (VLM), and an NSF Graduate Research Fellowship (SEG). This work was also supported in part by the Intramural Research Program of NINDS, NIH.

## Additional information

### Funding

| Funder | Grant reference number | Author |
|---|---|---|
| National Institutes of Health | K22 ES028048 | Catie Chang |

| National Institutes of Health | R01 NS112252 | Dario J Englot Catie Chang |
| National Institutes of Health | R01 NS108445 | Victoria L Morgan |
| National Institutes of Health | R01 NS110130 | Victoria L Morgan |
| National Science Foundation | Graduate Research Fellowship | Sarah E Goodale |

The funders had no role in study design, data collection and interpretation, or the decision to submit the work for publication.

## Author contributions
Sarah E Goodale, Conceptualization, Formal analysis, Investigation, Methodology, Writing - original draft, Writing - review and editing; Nafis Ahmed, Chong Zhao, Software, Investigation, Writing - review and editing; Jacco A de Zwart, Software, Investigation, Methodology, Writing - review and editing; Pinar S Özbay, Investigation, Writing - review and editing; Dante Picchioni, Methodology, Writing - review and editing; Jeff Duyn, Dario J Englot, Victoria L Morgan, Funding acquisition, Methodology, Writing - review and editing; Catie Chang, Conceptualization, Formal analysis, Supervision, Funding acquisition, Investigation, Methodology, Writing - original draft, Project administration, Writing - review and editing

## Author ORCIDs
Sarah E Goodale (iD) http://orcid.org/0000-0003-0460-6299
Nafis Ahmed (iD) http://orcid.org/0000-0002-5465-5729
Jacco A de Zwart (iD) http://orcid.org/0000-0001-8155-8185
Catie Chang (iD) https://orcid.org/0000-0003-1541-9579

## Ethics
Human subjects: All subjects provided written informed consent, and procedures were approved by the Institutional Review Boards of the National Institutes of Health (Protocol 00-N-0082) and Vanderbilt University (IRB #181540).

## Decision letter and Author response
Decision letter https://doi.org/10.7554/eLife.62376.sa1
Author response https://doi.org/10.7554/eLife.62376.sa2

# Additional files

## Supplementary files
• Supplementary file 1. Scan details. 'x' indicates whether a subject contributed an eyes-closed resting state (ECR) scan and/or an eyes-closed auditory task (ECT) scan. Some subjects did not complete both ECR and ECT scans.

• Supplementary file 2. Data subsets for cross-validation of template reproducibility. Resting- state scans and task scans were broken down into two non-overlapping subgroups so that one subject's rest and task scans did not cross into a different group.

• Supplementary file 3. Maximum frame-to-frame displacement for each subject's rest and task scans, as calculated during the motion coregistration pre-processing step.

• Transparent reporting form

## Data availability
Data files supporting the main findings are available on Open Science Framework, and code is available at https://github.com/neurdylab/fMRIAlertnessDetection (copy archived at https://archive.softwareheritage.org/swh:1:rev:db5f3ef49e53585412627984c44b9db9b188a868).

The following dataset was generated:

| Author(s) | Year | Dataset title | Dataset URL | Database and Identifier |
|---|---|---|---|---|
| Goodale SE, Ahmed N, Zhao C, de Zwart JA, Özbay PS, Picchioni D, Duyn JH, Englot DJ, Morgan VL, Chang C | 2020 | fMRI prediction of alertness and behavioral variability: data files | https://doi.org/10.17605/OSF.IO/MJ96B | Open Science Framework, 10.17605/OSF.IO/MJ96B |

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
