## [Decision Letter]

**Acceptance summary:**

This manuscript evaluates and extends a method for estimating levels of alertness in fMRI. The primary contributions include alertness prediction using a subset of voxels and the alertness measure's usage as a regressor in task fMRI. Reducing the required spatial map from the whole brain to a thresholded set of voxels is an interesting methodological contribution. Further, results suggest that this approach can help address alertness as a possible confound in task fMRI analyses.

**Decision letter after peer review:**

Thank you for submitting your article "fMRI-based detection of alertness predicts behavioral response variability" for consideration by *eLife*. Your article has been reviewed by 3 peer reviewers, one of whom is a member of our Board of Reviewing Editors, and the evaluation has been overseen by Christian Büchel as the Senior Editor. The reviewers have opted to remain anonymous.

The reviewers have discussed the reviews with one another and the Reviewing Editor has drafted this decision to help you prepare a revised submission.

Summary:

This manuscript evaluates multimodal fMRI-EEG data to estimate and assess attention in the fMRI signal based on its correlation with the EEG signal. The study of attention using fMRI is a well-studied topic, including somatosensory, visual, and auditory paradigms. Still, this study adds to the literature by using multimodal data to create the fMRI model. The discussed method follows prior work on the same approach and explores using a subset of voxels and applications as an fMRI signal regressor.

Essential revisions:

1. Conceptually, improved methods for addressing alertness confounds are a valuable contribution. The evaluation of the alertness index via the auditory task is a useful contribution. In terms of specific contributions, the reduction in the required spatial map, from the whole brain to a thresholded set of voxels, is an interesting methodological contribution. However, perhaps the main methodological contribution can be achieved more effectively using a direct approach, e.g., using a lasso model to predict the EEG alertness score using the fMRI time series. The lasso is considered a relatively standard statistical model, so it is not clear that the slight reduction in complexity is worth the loss in performance. At the very least, the authors should compare this approach to the presented thresholded correlation.

2. Reproducibility should be described more clearly. Reviewers recommend using cross-validation, dividing the data into 2 sets. The models can be built separately on each set, then compared. Also, one set's model can be used to evaluate prediction in the other set, and thus calculate sensitivity and specificity for this model. This would be complementary to the permutation testing already performed, as permutation testing is somewhat limited because these signals appear highly periodic (e.g., Figure 2) and thus can likely mismatch easily.

3. For the rsfMRI, participants were instructed to stay awake. Was this evaluated? There is a concern that participants might fall asleep during the 24.5m duration of the scan.

4. The auditory fMRI task made use of different tones. How were these tones delivered? Where earbuds used?

5. In the processing, it is stated that motion coregistration has been performed. Was there a quality check for excessive movement? When a participant moves more than the voxel size, the data are often discarded.

6. While testing hit/miss is clear, testing reaction times are limited to faster or slower than a single time (565 ms). As the study describes this method's use on an individual basis, consider intraindividual methods previously published, e.g., those from Thompson et al. 2013 (Citation 19), which includes separation by individual means, and also a separation of individuals into fast and slow groups.

7. A reaction time threshold of 565ms was used, but it remains unclear what the reason is for splitting the trails in an equal number of fast and slow trails. A reaction time of 565ms is still fast, and there can be a lot of variation between the fast and slow trails. A non-response was recorded if there was no button press within 5s. Therefore, perhaps fast can also be stated as below 2.5s and slow above 2.5s. The reference to Lim and Dinges (2008) indeed used a threshold at 0.5s but in a different setting.

8. A sample of 14 participants was investigated in this study. Would this sample be sufficient to prove the point of predicting fMRI results? There is no apriori sample size calculation, and therefore, it remains unknown if the data are sufficient to prove the point. In fMRI, usually larger samples up to 80 and more used to reduce variability/noise in the measurement.

9. The authors seem to indicate that the alertness metric could be used in resting-state fMRI data. Can't this be tested on either the authors' data or a publically available database? All that is needed is data with an index of alertness (such as eye-tracking) or even self-report of alertness/sleepiness.

10. Figure 5B shows positive (red) correspondence in a ventricle. This makes one wonder if this method is picking up on the EEG-linked CSF fluctuations reported elsewhere, e.g., Kiviniemi, V. et al. (2016). J Cereb Blood Flow Metab 36(6): 1033-1045. Can the authors discuss this?

11. What software was used for the processing of the data? Is the code available online? Please share via GitHub. Also, it remains unclear which statistical software was used for the statistical analyses.

[Editors' note: further revisions were suggested prior to acceptance, as described below.]

Thank you for submitting your article "fMRI-based detection of alertness predicts behavioral response variability" for consideration by *eLife*. Your article has been overseen by a Reviewing Editor and Christian Büchel as the Senior Editor.

The Reviewing Editor has drafted this to help you prepare a revised submission.

Essential revisions:

The reviewing editor believes that this manuscript is much improved and has addressed the conceptual and scientific concerns raised in the first round of reviews. While concerns about the sample size remain, the reviewing editor believes that this manuscript can be accepted as the contributions are primarily methodological.

The reviewing editor requests that the authors address the sample size limitations more directly in the main manuscript. In addition to the justification given in the response, consider adding language to the manuscript that clarifies that the findings may not generalize to the population from which they have sampled. Authors should also consider mentioning that their small sample likely overestimates the effect size of their results.

---

## [Author Response]

Essential revisions:1. Conceptually, improved methods for addressing alertness confounds are a valuable contribution. The evaluation of the alertness index via the auditory task is a useful contribution. In terms of specific contributions, the reduction in the required spatial map, from the whole brain to a thresholded set of voxels, is an interesting methodological contribution. However, perhaps the main methodological contribution can be achieved more effectively using a direct approach, e.g., using a lasso model to predict the EEG alertness score using the fMRI time series. The lasso is considered a relatively standard statistical model, so it is not clear that the slight reduction in complexity is worth the loss in performance. At the very least, the authors should compare this approach to the presented thresholded correlation.

The reviewers bring up a very interesting suggestion. To examine this point, we now compare the template-based approach to a lasso model. Just as we had done with the template method, here we train the lasso model on the resting-state fMRI scans and evaluate the performance on the task scans. For the lasso, we tested two models: one using a 268-region atlas (Shen et al., 2013) to parcellate the fMRI data into ROI time series as input to the model, and one using the fMRI time series from all voxels. For each approach, we trained a lasso model on the resting state data (temporally concatenated across subjects), using 10-fold cross-validation to choose the regularization parameter λ. We then applied the resulting coefficients (β weights) to the task fMRI data, to estimate an alertness time course for each task scan.

Figure 2—figure supplement 2 shows the correlation between the lasso-estimated fMRI arousal index and the EEG alertness index in each of the task scans. For comparison, correlations resulting from the fMRI alertness index estimated from the template-based approach are also shown. This figure shows that although our template method performed slightly better, the lasso model gave similar predictions, especially when using the 268-region atlas ROIs as input. This finding supports the notion that vigilance can be robustly estimated from fMRI data, and indicates that there may be multiple approaches for successfully extracting this information. Moreover, the initial feature reduction using a 268-region atlas seemed to improve the prediction beyond using a voxelwise lasso model. We believe that further investigating alternate linear models for vigilance prediction is an interesting area that we plan to develop in future work, including the use of other regularization methods such as ridge regression and elastic net, as well as alternate atlases for ROI definition. We now include these results as supplementary material, and have added the following to the Results (p.16), Discussion (p.22), and Methods (p. 33):

Results pg 16: “Finally, to investigate how this template-based method compares against a more conventional temporal regression approach, we also constructed lasso regression models (see Methods for further details). […] Further, the lasso model that was trained on fMRI time series from a 268-region atlas (Shen et al., 2013) was superior to that which used the time series from all voxels as input.”

Discussion p. 22: “Further, in this study we only considered a template-based approach and a lasso regression model to predict alertness. […] Further investigation of alternate linear models for vigilance prediction would be an interesting area to develop in future work, including the use of other regularization methods as well as parcellation atlases for deriving input time series.”

Methods p. 33: “While our primary analyses used a spatial template-based prediction approach, we also examine an alternate, temporal (lasso) regression model. Just as with the template method, here we train the lasso model on the resting-state fMRI scans, and evaluate the performance on the task scans. […] We then applied the resulting lasso coefficients to the task data, to estimate an alertness time course in each task scan.”

2. Reproducibility should be described more clearly. Reviewers recommend using cross-validation, dividing the data into 2 sets. The models can be built separately on each set, then compared. Also, one set's model can be used to evaluate prediction in the other set, and thus calculate sensitivity and specificity for this model. This would be complementary to the permutation testing already performed, as permutation testing is somewhat limited because these signals appear highly periodic (e.g., Figure 2) and thus can likely mismatch easily.

We agree that we could more clearly show the reproducibility of our results. First, we’d like to clarify that our permutation testing was carried out by shuffling across-scans, rather than shifting signals within scans, to avoid issues of periodicity. I.e., the EEG signals (entire time series) were randomly mismatched with the fMRI data of the other subjects. We have now clarified this in the manuscript on p.28. We would also like to highlight that our original analysis used a hold-out validation scheme, in which we trained the model (derived the template) from the set of resting-state scans and applied it to the task scans. However, although disjoint sets of scans were used for training and testing, some of the same subjects contributed both a resting-state and a task scan, as detailed in our original Supplementary Table (Supplementary File 1). To further probe the reliability and across-subject generalizability of our approach, we have now divided the subjects into two separate groups, and examine the performance when training/testing on all 4 possible combinations. As shown below, we find that these various train-test splits have similar performance. We have added this to our Results and Methods sections on p. 7 and p. 29-30, respectively:

Results: “To probe the across-subject reproducibility of fMRI alertness prediction within our dataset, we performed an additional cross-validation analysis in which subjects were divided into two distinct groups. Model training was performed on the resting-state data from one group, and evaluated on the task data of the other group. Cross-correlations between the estimated resting-state fMRI alertness index and the measured EEG alertness index were comparable to the original results (Figure 2—figure supplement 1).”

Methods: “To further investigate the reproducibility of fMRI alertness prediction, subjects were divided into two non-overlapping subgroups for both resting and task conditions (Supplementary File 2). […] Further, to test whether a model trained on task data can be used to predict vigilance in resting state, we also performed the reverse, constructing our template this time with the task scans from each subject group, and evaluating the fMRI alertness index on the held-out resting-state scans (Figure 2—figure supplement 3).”

3. For the rsfMRI, participants were instructed to stay awake. Was this evaluated? There is a concern that participants might fall asleep during the 24.5m duration of the scan.

Although the instruction was for subjects to stay awake, the long scans were designed to elicit variability in vigilance state – particularly, reduced wakefulness and drifts between wakefulness and light sleep – in order to estimate the fMRI pattern that tracks alertness fluctuations as well as evaluate its use for detecting alertness in situations during which the subject may be drowsy or lightly sleeping during a task. However, the point raised by the reviewers is important for several reasons: (i) characterizing sleep stage would give a clearer picture of the data from which our present results are generated, and (ii) it is an open question whether a template map derived from data that includes N1 or N2 sleep is just as effective as one derived from the wakeful but drowsy state.

Half of our resting state and task data have been independently sleep scored by two certified sleep techs at the Vanderbilt University Medical Center. Standard sleep scoring was carried out in 30-second epochs, according to standard AASM criteria. These ratings indicated that, across all of the staged scans combined, 85.6% of epochs were rated as ‘awake’, suggesting that subjects were mostly alert. On pg. 24 in Methods, we now include:

“Although the instruction was for subjects to stay awake, the long scans were designed to elicit variability in vigilance state – particularly, reduced wakefulness and drifts between wakefulness and light sleep. […] It is therefore likely state sleep was not a major factor in our experiments”.

4. The auditory fMRI task made use of different tones. How were these tones delivered? Where earbuds used?

We have added this information into our Methods (p. 24): “Auditory tones were delivered via earbuds placed over subjects’ ear plugs.”

5. In the processing, it is stated that motion coregistration has been performed. Was there a quality check for excessive movement? When a participant moves more than the voxel size, the data are often discarded.

We agree it is important to consider head motion. We first note that our data is collected via multi-echo acquisition, and the multi-echo ICA denoising approach described in Kundu et al. (2012, 2013) was applied. This approach has been demonstrated to be very effective in reducing head motion effects (Kundu et al., PNAS 2013) by isolating BOLD from non-BOLD components based on the echo-time dependence of fMRI percent signal change. However, we have now also performed quality checks for excessive movement. For each scan, we have calculated the maximum frame-to-frame displacement, finding that only two subjects had a maximum displacement that exceeded our voxel size (3 mm), see Supplementary File 3. We additionally find that removing these subjects had no effect on the main results, (Figure 4—figure supplement 4). This has now been added into the Methods section (p.25) and Results (p. 12):

p. 25: “Maximum frame-to-frame displacement was also calculated, as shown in Supplementary File 3.”

p. 12: “Finally, while multi-echo ICA has proven effective in reducing head-motion effects (Kundu et al., PNAS 2013), we also ensured that scans with frame-to-frame head motion exceeding our voxel size (3 mm; corresponding to 1 resting-state and 1 task scan; Supplementary File 3) did not impact the major findings of this study. Removing these high-motion scans from the analysis had minimal impact on the correspondence between the fMRI alertness index with EEG and behavior (Figure 4—figure supplement 4).”

6. While testing hit/miss is clear, testing reaction times are limited to faster or slower than a single time (565 ms). As the study describes this method's use on an individual basis, consider intraindividual methods previously published, e.g., those from Thompson et al. 2013 (Citation 19), which includes separation by individual means, and also a separation of individuals into fast and slow groups.

We thank the reviewer for this suggestion. We have updated our explanation of our threshold selection in our methods which was found by taking the median of all trials. Our use of 565ms as the threshold came from the median across all pooled reaction times. However, we agree that it is also interesting to look at intra-individual thresholds. So now, for each scan, we also took the fastest and slowest 10% of trials (as has been used in psychomotor vigilance tasks; Loh et al., 2004), and report those results in Figure 4—figure supplement 1.

The paired t-test was still significant. One difference in the bar plots (compared to Figure 4B-C) is that the ‘slow’ (yellow) alertness index is closer to that of the ‘miss’ (blue). This would make sense, given that this analysis considers the slowest 10% of trials; this pattern is also evident in the EEG, lending further confidence to our result. We also considered a between-subjects split (i.e., assigning entire subjects into a ‘fast’ or ‘slow’ group based on their median RT), but failed to find a significant separation (p>0.05, two-sample t-test). One potential reason is because the variance of reaction times within some subjects were quite large.

We have now described the new findings in Results p.9-10 and Methods p.31 of the revised manuscript.

7. A reaction time threshold of 565ms was used, but it remains unclear what the reason is for splitting the trails in an equal number of fast and slow trails. A reaction time of 565ms is still fast, and there can be a lot of variation between the fast and slow trails. A non-response was recorded if there was no button press within 5s. Therefore, perhaps fast can also be stated as below 2.5s and slow above 2.5s. The reference to Lim and Dinges (2008) indeed used a threshold at 0.5s but in a different setting.

We’d like to reference essential revision 6 for the first part of this comment. We have considered that there are alternate ways to find the threshold and have now also performed an intra-individual separation into fast and slow trials. With regard to the 2.5s threshold, we have now also considered this, but find that across all subjects and trials, there was only one reaction time slower than 2.5s (histogram of reaction times Figure 4—figure supplement 5; now described in the Methods on p. 31). Thus, it appears that our alertness index can discriminate between slow/fast reaction times even amongst a set of relatively fast reaction times, supporting its sensitivity as a marker of underlying brain state.

8. A sample of 14 participants was investigated in this study. Would this sample be sufficient to prove the point of predicting fMRI results? There is no apriori sample size calculation, and therefore, it remains unknown if the data are sufficient to prove the point. In fMRI, usually larger samples up to 80 and more used to reduce variability/noise in the measurement.

We acknowledge the reviewers’ concern about sample size, and have now pointed this out as a limitation in the Discussion. Nonetheless, there are several factors that increase our confidence in the present results. First, our scans are long (24.5 min per scan) and 310 total trials were available for mining behavioral responses. Second, the additional analyses suggested by the reviewers (see Comment 2 above) also support the generalization ability of the approach. Third, we also note that our subject sample size is on par with other multi-modal (EEG-fMRI) studies that have examined vigilance effects in fMRI and, importantly, whose results have since been replicated in independent studies. These include Falahpour et al., 2018 (10 subjects, used to investigate the same template-based vigilance prediction on resting-state fMRI; similar effects are replicated in our current study), and Olbrich et al., 2009 (15 subjects, used to evaluate fMRI activity associated with vigilance stage changes [with similar effects replicated in Ong et al., 2015]). Similarly, Hesselmann et al. 2008 used 12 subjects to study the ability to predict behavior from pre-trial fMRI activity, and Poudel et al. 2014 used a subset of 14 subjects to map activity associated with microsleeps. Regardless, we do agree that our subject size is a limitation and we are looking to expand on this for future studies. We have added this into the Discussion on pg. 22:

“Another limitation of this study is the relatively small sample size, and future work will be carried out with this approach using larger sample sizes as well as other experimental conditions. Nonetheless, there are several factors that increase our confidence in the present results: our scans are long (24.5 min per scan), and the replication analyses carried out in Figure 2—figure supplement 1 and Figure 4—figure supplements 1 and 4 further support the generalizability of this approach.”

9. The authors seem to indicate that the alertness metric could be used in resting-state fMRI data. Can't this be tested on either the authors' data or a publically available database? All that is needed is data with an index of alertness (such as eye-tracking) or even self-report of alertness/sleepiness.

We have now tried to clarify in this paper that this alertness metric has been successfully applied to resting-state data in previous work (Falahpour et al., 2018 in humans; Chang et al., 2016 in macaques). The main goal of the present study was to investigate the ability of this alertness metric to predict behavioral response variability throughout a task, and at a fine temporal resolution (i.e., looking in a 5s pre-stimulus window before the trial). In doing so, our resting-state scans served as a “training set” from which we created the template, and all evaluation was applied to the held-out (task) scans. However, as the reviewers point out, we can also examine the performance on resting-state fMRI using the current dataset. We have now provided an additional analysis in which we construct the template with the task scans and evaluate the alertness prediction on the held-out resting-state scans (see reply to Comment 2 above, and Figure 2—figure supplement 3). In addition, another recent study (Gu et al., 2020) used the same approach independently to calculate vigilance in resting-state Human Connectome Project data, and found it agreed well with self-reported sleepiness levels. We have added this to pg. 22:

“Further, although the present study centered on predicting task-related behavioral responses, we also performed an analysis in which the template was constructed from the task scans, and the fMRI alertness index was evaluated on the held-out resting-state scans, obtaining comparable results (Figure 2—figure supplement 3).”

10. Figure 5B shows positive (red) correspondence in a ventricle. This makes one wonder if this method is picking up on the EEG-linked CSF fluctuations reported elsewhere, e.g., Kiviniemi, V. et al. (2016). J Cereb Blood Flow Metab 36(6): 1033-1045. Can the authors discuss this?

Thank you for this comment, we agree it would be great to discuss this point. In experiments with widespread BOLD changes, we have often seen an antipolar change in or near CSF, and in deep white matter (especially periventricularly). One potential mechanism is that these changes in CSF signals are at least in part due to changes in CBV, where CBV increases result in CSF volume decreases or vice versa (e.g., Ozbay, P.S. Neuroimage 2018, 176: 541). This is expected in peri-venous CSF spaces that are abundant in peri-ventricular white matter, within the ventricles near choroid plexus. In these regions, signal changes associated with CSF volume changes may substantially precede (and outweigh) BOLD effects. It has been reported that conditions that lead to vasoaction also change the EEG baseline shifts (Nita D.A. J Neurophysiol 2004, 92:1011), and this has also been suggested in the work of Kiviniemi et al. One difference between our study and the work of Kiviniemi et al. is that the latter used DC EEG data, whereas our fMRI arousal template was based on EEG band limited power, which effectively discarded the DC signal. Therefore, we believe our template may be picking up other mechanisms, but future work is necessary to examine this in more detail.

We have added a brief discussion of this possibility on p. 19.

“A plausable mechanistic explanation offered recently is that widespread fMRI signal changes generated by either neuronal or autonomic activity lead to antagonistic fMRI signal changes in the downstream venous vasculature, in part due to a disjoint timing and amplitude relationship of blood volume and blood oxygenation effects^56^. Further work will be necessary to investigate potential links between these observations and those of studies which have observed CSF fMRI signals in other EEG frequencies (Kiviniemi et al., 2016). ”

11. What software was used for the processing of the data? Is the code available online? Please share via GitHub. Also, it remains unclear which statistical software was used for the statistical analyses.

We have updated our methods to specify the software for processing and statistical analyses. All analyses were performed in MATLAB (R2018b). This has been updated on p. 34. The code is now available via GitHub, and a link is now provided in the Data Availability field in the submission site.

References:

1. Shen X, Tokoglu F, Papademetris X, Constable RT. Groupwise whole-brain parcellation from resting-state fMRI data for network node identification. Neuroimage. 2013 Nov 15;82:403-15. doi: 10.1016/j.neuroimage.2013.05.081.

2. American Academy of Sleep Medicine. (American Academy of Sleep Medicine, Westchester, IL, 2007).

3. Kundu, P., Inati, S. J., Evans, J. W., Luh, W.-M. and Bandettini, P. A. Differentiating BOLD and non-BOLD signals in fMRI time series using multi-echo EPI. Neuroimage 60, 1759-1770, doi:10.1016/j.neuroimage.2011.12.028 (2012).

4. Kundu, P. et al. Integrated strategy for improving functional connectivity mapping using multiecho fMRI. Proc. Natl. Acad. Sci. U. S. A. 110, 16187-16192, doi:10.1073/pnas.1301725110 (2013).

5. Thompson, G. J. et al. Short-time windows of correlation between large-scale functional brain networks predict vigilance intraindividually and interindividually. Human Brain Mapping 34, 3280-3298, doi:10.1002/hbm.22140 (2013).

6. Loh, Sylvia, et al. "The validity of psychomotor vigilance tasks of less than 10-minute duration." Behavior Research Methods, Instruments, and Computers 36.2 (2004): 339-346.

7. Falahpour, M., Chang, C., Wong, C. W. and Liu, T. T. Template-based prediction of vigilance fluctuations in resting-state fMRI. Neuroimage 174, 317-327, doi:10.1016/j.neuroimage.2018.03.012 (2018).

8. Olbrich, S. et al. EEG-vigilance and BOLD effect during simultaneous EEG/fMRI measurement. Neuroimage 45, 319-332, doi:10.1016/j.neuroimage.2008.11.014 (2009).

9. Ong, Ju Lynn, et al. "Co-activated yet disconnected—Neural correlates of eye closures when trying to stay awake." Neuroimage 118 (2015): 553-562.

10. Hesselmann, G., Kell, C. A., Eger, E. and Kleinschmidt, A. Spontaneous local variations in ongoing neural activity bias perceptual decisions. Proc. Natl. Acad. Sci. U. S. A. 105, 10984-10989, doi:10.1073/pnas.0712043105 (2008).

11. Poudel, G. R., Innes, C. R. H., Bones, P. J., Watts, R. and Jones, R. D. Losing the struggle to stay awake: divergent thalamic and cortical activity during microsleeps. Hum. Brain Mapp. 35, 257-269, doi:10.1002/hbm.22178 (2014).

12. Chang, C. et al. Tracking brain arousal fluctuations with fMRI. Proc. Natl. Acad. Sci. U. S. A. 113, 4518-4523, doi:10.1073/pnas.1520613113 (2016).

13. Gu, Y., Han, F., Sainburg, L. E., and Liu, X. (2019). Transient arousal modulations are responsible for resting-state functional connectivity changes associated with head motion. bioRxiv, 444463.

14. Kiviniemi, V., Wang, X., Korhonen, V., Keinänen, T., Tuovinen, T., Autio, J.,.… and Nedergaard, M. (2016). Ultra-fast magnetic resonance encephalography of physiological brain activity–glymphatic pulsation mechanisms?. Journal of Cerebral Blood Flow and Metabolism, 36(6), 1033-1045.

[Editors' note: further revisions were suggested prior to acceptance, as described below.]

Essential revisions:The reviewing editor believes that this manuscript is much improved and has addressed the conceptual and scientific concerns raised in the first round of reviews. While concerns about the sample size remain, the reviewing editor believes that this manuscript can be accepted as the contributions are primarily methodological.The reviewing editor requests that the authors address the sample size limitations more directly in the main manuscript. In addition to the justification given in the response, consider adding language to the manuscript that clarifies that the findings may not generalize to the population from which they have sampled. Authors should also consider mentioning that their small sample likely overestimates the effect size of their results.

We agree that it would be helpful to address this point more directly in the manuscript. We now mention both points suggested by the Reviewing Editor: (1) that findings may not generalize to the population; (2) the small sample size may have led to overestimation of effect size. This material is now included on p. 22 of the revised submission:

“One key limitation of this study is the small sample size. As a result, the effect sizes of the current results may be overestimated, and findings may not generalize well to the population from which they are sampled. Future work should be carried out with this approach using larger sample sizes as well as other experimental conditions.”

Additional note: During the last review period, we had also come across 3 relevant references that we also wish to cite. The added references are listed below, and have been cited on p.3 and 19:

4. Makeig, S., Jung, T. P. and Sejnowski, T. J. Awareness during drowsiness: dynamics and electrophysiological correlates. Can J Exp Psychol 54, 266-273 (2000).

5. Poudel, G. R., Innes, C. R. H. and Jones, R. D. Temporal evolution of neural activity and connectivity during microsleeps when rested and following sleep restriction. Neuroimage 174, 263-273, doi:10.1016/j.neuroimage.2018.03.031 (2018).

57. McAvoy, M. P., Tagliazucchi, E., Laufs, H. and Raichle, M. E. Human non-REM sleep and the mean global BOLD signal. J Cereb Blood Flow Metab 39, 2210-2222, doi:10.1177/0271678X18791070 (2019).